# Using the Decomposition-Based Multi-Objective Evolutionary Algorithm with Adaptive Neighborhood Sizes and Dynamic Constraint Strategies to Retrieve Atmospheric Ducts

**DOI:** 10.3390/s20082230

**Published:** 2020-04-15

**Authors:** Yanbo Mai, Hanqing Shi, Qixiang Liao, Zheng Sheng, Shuai Zhao, Qingjian Ni, Wei Zhang

**Affiliations:** 1College of Meteorology and Oceanography, National University of Defense Technology, Nanjing 210000, China; maiyanbo163@163.com (Y.M.); liaoqixiang2013@126.com (Q.L.); 19994035@sina.com (Z.S.); zhangwei19@nudt.edu.cn (W.Z.); 2Collaborative Innovation Center on Forecast and Evaluation of Meteorological Disasters, Nanjing University of Information Science and Technology, Nanjing 210000, China; 3School of Computer Science and Engineering, Southeast University, Nanjing 210000, China; 220181728@seu.edu.cn (S.Z.); nqj@seu.edu.cn (Q.N.)

**Keywords:** special sensor, GNSS, new algorithm, balance the diversity and convergence of the population, atmospheric ducts

## Abstract

The traditional method of retrieving atmospheric ducts is to use the special sensor of weather balloons or rocket soundings to obtain information intelligently, and it is very expensive. Today, with the development of technology, it is very convenient to retrieve the atmospheric ducts from Global Navigation Satellite System (GNSS) phase delay and propagation loss observation data, and then the GNSS receiver on the ground forms an automatic receiving sensor. This paper proposes a hybrid decomposition-based multi-objective evolutionary algorithm with adaptive neighborhood sizes (EN-MOEA/ACD-NS), which dynamically imposes some constraints on the objectives. The decomposition-based multi-objective evolutionary algorithm (MOEA/D) updates the solutions through neighboring objectives, the number of which affects the quality of the optimal solution. Properly constraining the optimization objectives can effectively balance the diversity and convergence of the population. The experimental results from the Congress on Evolutionary Computation (CEC) 2009 on test instances with hypervolume (HV), inverted generational distance (IGD), and average Hausdorff distance ∆_2_ metrics show that the new method performs similarly to the evolutionary algorithm MOEA/ACD-NS, which considers only the dynamic change of the neighborhood sizes. The improved algorithm is applied to the practical problem of jointly retrieving atmospheric ducts with GNSS signals, and its performance further demonstrates its feasibility and practicability.

## 1. Introduction

In both technical practice and theoretical research, there are a large number of multi-objective optimization (MOP) problems that need to be optimized synchronously for multiple objectives [1,2,3]. When solving multi-objective optimization problems, the traditional method is to convert this problem into a single-objective problem based on prior knowledge and to use single-objective optimization methods to obtain a satisfactory Pareto optimal solution [4]. The Pareto solution contains a lot of information, which can be used to analyze the correlation between multiple objectives; furthermore, the multi-objective optimization can make full use of multi-source observation information and can improve the reliability of the solution. The traditional method can only obtain a Pareto optimal solution that is not necessarily satisfactory to the decision-maker; the rationality of prior knowledge also affects the selection of the optimal solution. This paper proposes a multi-objective evolutionary algorithm (MOEA) for MOP problems in order to obtain a set of Pareto optimal solutions [5,6]. According to the optimization framework, MOEAs can be roughly divided into three categories, namely: (a) Pareto-based approaches [7], (b) indicator-based approaches [8], and (c) decomposition-based approaches [9].

Pareto-based methods sort the individuals according to the dominant relationship among them, and then the optimal solution is selected. The Pareto-based approach is the most popular method in the evolutionary multi-objective optimization community. NSGA-II and SPEA2 are the most representative algorithms, and they are very effective when dealing with two or three objective functions [7,10]. However, when the number of objective functions increases to more than three, most of the solutions in the NSGA-II and SPEA2 search spaces become non-dominated solutions, resulting in a rapid decline in search capabilities.

Indicator-based MOEAs use the value of performance indicators to guide the search direction, which drives whole generated solutions to the Pareto solution sets. As the calculation of the Pareto dominance relationship is not involved, the computational complexity is reduced. The most commonly used indicators are hypervolume (HV) and R2, which provide the basis for the representative algorithms, SMS-MOEA and SIBEA [8,11]. The main drawback of these algorithms is the heavy indicator calculation. In particular, the calculation time increases exponentially with increases in the solution dimension when dealing with a multi-objective optimization problem. Although researchers have tried to reduce computational costs and replace the indicator calculation with other evaluation metrics, such as ∆_p_, these improved algorithms are still unsatisfactory [12,13].

Decomposition-based approaches decompose the multi-objective function into several scalar functions through aggregation functions and optimize each objective in parallel with some effective evolutionary algorithms. The diversity of the population is controlled by the weight vector, which indicates each objective’s importance. The decomposition-based multi-objective evolutionary algorithm (MOEA/D) and multiple single-objective Pareto sampling algorithms (MSOPS) are the most representative algorithms [9]. In the MOEA/D framework, there are several aggregation functions, such as weighted sum (WS), weighted Tchebycheff (WT), and penalty-based boundary intersection (PBI) [14]. After the objective function is decomposed, the solution of an objective is optimized mainly through its neighboring objectives, so that the neighborhood sizes (NS) naturally affect the quality of the whole optimal solution set. In order to improve the MOEA/D, Zhao et al. proposed a method called an ensemble of different neighborhood sizes with online self-adaptation (ENS-MOEA/D), which aims to avoid the local optimum or poor convergence due to the neighborhood sizes [15]. Martín et al. developed a novel predictor-corrector method with the aim of finding the zero set of F-an underdetermined system of equations motivated by the Karush–Kuhn–Tucker conditions for MOP problems [16]. Based on the variable neighborhood tabu search, Janssens et al. proposed a method to analyze the effect of the algorithmic parameters and instance characteristics on the quality of a Pareto front produced by a multi-objective algorithm [17]. Yang et al. developed a new variant of MOEA/D with a dual-information and dual-selection (DS) strategy (MOEA/D-DIDS), which used an adaptive historical and neighboring information for generating new individuals in order to avoid local optima and accelerate the convergence rate [18]. Furthermore, many mathematical methods, such as cell mapping methods, subdivision techniques, and continuation methods, have been applied to multi-objective optimization problems, which provide more references for solving multi-objective optimization problems in different fields. Cell mapping methods can detect the global Pareto set in one run of the algorithm, but they are restricted to small dimensional problems, as the number of cells grows exponentially with the number of dimensions [19]. Subdivision techniques can approximate the entire set of global Pareto points within a compact domain, but the computational time is quite large in the higher- dimensional parameter space [20]. Continuation methods can be applied very efficiently to perform the search along the Pareto set, even for high-dimensional models but are of local nature [21]. However, there are still some defects in their algorithm. If the evolution regions are too large, a single new solution may replace several old solutions and further diminish the population diversity. Based on the ENS-MOEA/D framework, and in order to balance the population diversity and convergence, this paper combines the constrained decomposition approach with the adaptive strategy proposed by Wang et al. to reduce the improvement regions and control the evolution direction [22] and apply it to solve the problem of the joint inversion of atmospheric ducts based on global navigation satellite system (GNSS) signals.

The propagation of an electromagnetic wave in the atmospheric boundary layer, especially in the surface layer, is affected by atmospheric refraction, and the propagation path will bend to the ground. Under certain meteorological conditions, the curvature of the path may exceed the curvature of the Earth’s surface, and the electromagnetic wave will be partially trapped in a thin layer of atmosphere with a certain thickness. This phenomenon is called the atmospheric ducts [23]. The atmospheric ducts can capture an electromagnetic wave and change its propagation path, so that the electromagnetic wave can be propagated beyond the horizontal sight distance of the electromagnetic system, realizing the over sight distance detection and over sight distance reception. Therefore, it is very important to study the atmospheric ducts for electromagnetic communication, and their influence on the propagation of electromagnetic wave has a broad application prospect. So far, with the development of computers, more and more intelligent algorithms are being applied in the field of meteorology [24,25,26,27,28,29]. The traditional method of retrieving atmospheric ducts is to use the special sensor of weather balloons or sounding rockets to obtain information intelligently; however, weather balloons and sounding rockets are expensive in terms of expendables and manpower. Today, with the development of technology, it is very convenient to obtain atmospheric ducts from GNSS phase delay and propagation loss observation data [30], so we tried to retrieve the atmospheric ducts by placing a receiver on the ground to receive the signal from the GNSS and use a new algorithm to analyze it. This method will would save a lot of costs in the retrieval of atmospheric ducts.

In this paper, we propose a hybrid decomposition-based multi-objective evolutionary algorithm with adaptive neighborhood sizes (EN-MOEA/ACD-NS), which dynamically imposes some constraints on the objectives, and compare the improved algorithm with the MOEA/D, MOEA/ACD, MOEA/ACD-NS, and EN-MOEA/D on the MOPs and unconstrained function (UF) problems in the Congress on Evolutionary Computation (CEC) 2009 standard test instances [31]. To demonstrate the practical application of our algorithm and solve the practical problem of jointly retrieving atmospheric ducts, we applied the algorithm to solve the problem of the joint retrieval of atmospheric ducts based on GNSS signals [32]. This new algorithm will provide a new method and save a lot of costs for the retrieval of atmospheric ducts, which is of great significance for the research of atmospheric ducts.

## 2. Basic Concepts of Multi-Objective Optimization and Evaluation Metrics

### 2.1. Definition of Multi-Objective Optimization

Some problems that we encounter in life have only one objective function and are known as single-objective problems, such as the classic traveling salesman problem (TSP) problem. Some problems, called multi-objective problems, have more than two objective functions whose goals are contradictory, such as logic circuit design, and it is difficult for each function to achieve optimal solutions at the same time [33]. A problem with *m* optimization goals and *n* decision variables can be expressed as follows [32,34]:(1)$$min:y=F\left(\vec{x}\right)={\left(f_{1}\left(\vec{x}\right),f_{2}\left(\vec{x}\right),\ldots ,f_{m}\left(\vec{x}\right)\right)}^{T}.$$

Here, $\vec{x}\in R^{n}$ represents the *n*-dimensional decision space and $f_{i}\left(\vec{x}\right)$ is the *i*th objective. The *k* inequality constraints and *l* equality constraints are the conditions constraining the above formula, which satisfy the following:(2)$$g_{i}\left(\vec{x}\right)\leq 0,\;i=1,2,\ldots ,k,$$
(3)$$h_{j}\left(\vec{x}\right)=0,\;i=1,2,\ldots ,l.$$

The objectives cannot be optimized synchronously. In order to achieve an optimal solution of the function $F\left(\vec{x}\right)$, the objectives that conflict with each other are considered comprehensively to meet the constraint conditions and the objective functions are optimized. Generally, the solution that satisfies the constraints is defined as a feasible solution. In order to meet the optimal conditions, Pareto dominance is introduced to describe the relationship between two feasible solutions, and then the Pareto optimal solution is found. Suppose there are two feasible solutions, ${\vec{x}}_{a}$ and ${\vec{x}}_{b}$, then ${\vec{x}}_{a}$ is called the Pareto dominant over ${\vec{x}}_{b}$ if $f_{i}\left({\vec{x}}_{a}\right)\leq f_{i}\left({\vec{x}}_{b}\right)$, i
∈$\left\{1,2,\ldots ,m\right\}$ for every $i\in \left\{1,2,\ldots ,m\right\}$ and $f_{j}\left({\vec{x}}_{a}\right)<f_{j}\left({\vec{x}}_{b}\right)$ for at least one index $j\in \left\{1,2,\ldots ,m\right\}$. If there is no other feasible solution ${\vec{x}}^{*}$ that are Pareto dominant over ${\vec{x}}_{a}$, then ${\vec{x}}_{a}$ is called the Pareto optimal dominating solution. The set composed of such solutions is the Pareto optimal solution set (PS), and the edge of the target space mapped by PS is called the Pareto front (PF) [4].

### 2.2. Evaluation Metrics

It is a very important and difficult task to evaluate the performance of MOEAs. At present, evaluation focuses mainly on the following three aspects: (a) the approximation degree between the solution set and the Pareto optimal surface (convergence), (b) the distribution uniformity of the solution set in the target space (distribution uniformity), and (c) the extensive degree of the solution set (widespread distribution) [14,31,35]. In this paper, the performance of the improved algorithm in different test problems is evaluated by HV, inverted generational distance (IGD), and the average Hausdorff distance ($\Delta_{2}$).

### 2.3. Hypervolume

HV was first proposed by Zitzler in 1998, and is called a set of quality test indicators of the “space coverage area” [36]. It is widely used for quantitative measurement of population quality because HV is still sensitive to Pareto dominance and population diversity in high dimensions and does not decrease with increasing dimensions. In general, HV is used to measure the hypervolume between non-dominated points and reference points, which can be expressed as follows [36]:(4)$$\mathrm{HV}=vol\left(\cup_{i=1}^{\left|p^{*}\right|}\left[p_{1},r_{1}\right]\times \left[p_{2},r_{2}\right]\times \cdots \times \left[p_{n},r_{n}\right]\right)HV=vol\left(\cup_{i=1}^{\left|p^{*}\right|}\left[p_{1},r_{1}\right]\times \left[p_{2},r_{2}\right]\times \cdots \times \left[p_{n},r_{n}\right]\right).$$

Here, $p^{*}=\left\{p_{1},p_{2},\ldots ,p_{n}\right\}$, $\left|p^{*}\right|$ is the number of reference points, and vol is the Lebesgue metric; the larger the value of HV, the better the performance of the algorithm.

### 2.4. Inverted Generational Distance

IGD is a comprehensive performance evaluation index. It mainly evaluates the convergence and distribution performance of the algorithm by calculating the sum of the minimum distance between each point (individual) on the real Pareto front and the group of individuals obtained by the algorithm. It can be expressed as follows [37]:(5)$$IGD\left(p^{*},Q\right)=\frac{\sum_{v\in P}d\left(v,Q\right)}{\left|p^{*}\right|}.$$

Here, $p^{*}=\left\{p_{1},p_{2},\ldots ,p_{n}\right\}$, $\left|p^{*}\right|$ is the number of reference points, and *Q* is the optimal Pareto solution set obtained by the algorithm. *D* (*v*, *Q)* is the minimum Euclidean distance between individual *v* and group *Q* in *P^*^*; the smaller the value of IGD, the better the performance of the algorithm.

### 2.5. Average Hausdorff Distance

The classical Hausdorff distance is a measurement criterion of the maximum and minimum distance, and is used to describe the similarity between two sets of points. However, it is susceptible to external interference, resulting in large errors in the final results. As a supplementary measure, Schütze et al. revised the original generational distance (GD) and IGD and proposed an average Hausdorff distance based on these two indicators [12]. Assuming that there is a set of non-dominated points $X=\left\{x_{1},x_{2},\ldots ,x_{n}\right\}$ and a set of reference points $P^{*}=\left\{p_{1},p_{2},\ldots ,p_{m}\right\}$ which is a finite non-empty set. Then, the Euclidean norm of the average Hausdorff distance is defined as follows [12]:(6)$$\Delta_{2}\left(X,P^{*}\right)=max\left({\left(\frac{1}{n}\sum_{i=1}^{n}dist{\left(x_{i},P^{*}\right)}^{2}\right)}^{1/2},{\left(\frac{1}{m}\sum_{i=1}^{m}dist{\left(X,p_{i}\right)}^{2}\right)}^{1/2}\right).$$

The smaller the value of $\Delta_{2}$, the better the performance of the algorithm.

## 3. MOEA/D Framework and Improvement

### 3.1. Basic Framework

The main idea of MOEA/D is to use an aggregate function to decompose a MOP problem into multiple single-objective optimization problems. The individual uses the information of adjacent solutions to evolve in order to obtain the optimal solution. At present, the improvement of MOEA/D is mainly concerned with improvements in the weight vector, decomposition method, matching selection, evolutionary operator, replacement and selection method, and computing resource allocation [38,39,40,41,42,43,44]. Various improvement methods have been carried out according to the basic framework of MOEA/D. The algorithm in this paper improves the decomposition method and matching selection. In the basic framework, some information needs to be saved each time, such as the solution set $\left\{x_{1},x_{2},\ldots ,x_{p}\right\}$ in which population size is *p*, the corresponding fitness set $\left\{FV_{1},FV_{2},\ldots ,FV_{p}\right\}$, the current optimal solution set $\left\{b_{1},b_{2},\ldots ,b_{p}\right\}$, and the external population EP. Here, $x_{i}$ (*I* = 1, 2, …, *p*) is the current solution to the *i*th objective. The basic framework can be expressed as follows [31,39]:

*Step 1: Initiation*.

Step 1.1: Let EP = Ø

Step 1.2: Calculate the Euclidean distance between two weight vectors according to which we select *T* individuals as a neighbor population. Set $B\left(i\right)=\left\{i_{1},i_{2},\ldots ,i_{T}\right\}$ for *i* = 1,2,...,*n*, and $\lambda_{j}$ as the neighbor vector to$\;\lambda_{i}$
$\forall j\in B\left(i\right)$.

Step 1.3: Generate an initial population randomly.

Step 1.4: Calculate $FV_{i}=F\left(x_{i}\right)$ for i=1,2,…,N.

Step 1.5: Set $B={\left\{b_{1},b_{2},\ldots ,b_{p}\right\}}^{T}$ according to the different problems.


*Step 2: Update.*


Step 2.1: After generating new generation offspring, select two individuals $x_{l}$ and $x_{k}$ from the adjacent set $B\left(i\right)$, and generate new children $x^{new}$ by a genetic algorithm or differential evolution [45].

Step 2.2: Adjust the $x^{new}$ by using a modified heuristic method for the particular problem.

Step 2.3: Stop loop if the stop condition is satisfied and output EP; otherwise, jump to Step 2.

### 3.2. Improvement

#### 3.2.1. Adaptive Neighborhood Sizes

The new solution generated by MOEA/D involves a number of neighboring objectives. The larger the number, the more likely the optimal individual is to be globally optimal. Otherwise, the smaller the number, the more likely it is to be locally optimal. Therefore, the determination of the number of neighboring problems (*T*) is related to the performance of the algorithm. Zhao et al. proposed a method to dynamically adjust the number of neighboring objectives [15]. This paper uses their methods to improve the original MOEA/D. First, we select several number values as candidates for the neighborhood sizes, such as$\;\left\{2,4,8,12,16\right\}$. The initial value is set to 2, and, when the number of iterations reaches a certain limit, such as 50, it is determined by probability $p_{k,G}$ whether the *T* is redefined. $p_{k,G}$ is expressed as follows:(7)$$p_{k,G}=\frac{S_{k,G}}{\sum_{i=1}^{K}S_{k,G}},$$
where
(8)$$S_{k,G}=\frac{\sum_{g=G-LP}^{G-1}TNS\_success_{k,g}}{\sum_{g=G-LP}^{G-1}TNS_{k,g}}+\delta ,\;k=1,2,\ldots ,K,G>LP.$$

$S_{k,G}$ represents the probability of successfully improving progeny when the neighboring number is *k* in the learning period (*LP*). $TNS_{k,g}$ is the number of new individuals generated by the children determined by the *k*th neighborhood size (NS). $TNS\_success_{k,g}$ is the number of individuals that have been successfully improved from $TNS_{k,g}$. Here, δ=0.05 is a small constant used to avoid $S_{k,G}$ becoming zero. 

#### 3.2.2. Adaptive Constraints Approach

In the previous section, dynamically adjusting NS only affects the number of fathers that may produce new individuals. However, how to select right individuals from parents to mate more quickly and efficiently is still a question that needs to be solved. Using the method of Reference [22], we could dynamically adjust the direction of paternal mating to further improve MOEA/D. After the calculations of the evolutionary algorithm, many new individuals will be generated. In order to reduce the distribution space of new individuals, the optimization direction should be limited. For example, in the *i*th objective, it must satisfy the following:(9)$$g\left(x\left|\lambda_{i},z^{*}\right.\right)=\min!\;,$$
where the constraint condition is $a_{i},F\left(x\right)-z^{*}\leq 0.5\theta_{i}$, x∈Ω.

$g\left(\cdot \right)$ is the aggregated objective function of each optimization objective determined by the weight vector $\lambda_{i}$; $z^{*}$ is a utopian point; $a_{i},F\left(x\right)-z^{*}$ is the acute angle between $a_{i}$ and $F\left(x\right)-z^{*}$, which represents the basic optimized direction; and $a_{i}$ is the optimized direction vector of the objective. Additionally, let $\theta_{i}$ be the control parameter that determines the size and direction of the optimizable region. Consequently, dynamically adjusting $\theta_{i}$ means controlling the optimized direction and optimizable population.

The specific dynamic strategy is as follows. Assuming that $x_{i}$ is the solution set of the current objective, and defining the divergence function $\alpha \left(x_{i}\right)=a_{i},F\left(x_{i}\right)-z^{*}$, then the solution $x_{i}$ is close to $z_{i}$ if $\alpha \left(x_{i}\right)$ is very small. At this point, the number of neighbors will increase. When all $\alpha \left(x_{i}\right)$ are small, the solution set *x* is close to the real value and the population has good diversity. The method of Reference [22] also provides another measurement: the relative divergence degree $\alpha_{r}\left(i\right)$, which is defined as $\alpha_{r}\left(x_{i}\right)=\left[n\cdot \alpha \left(x_{i}\right)\right]/\sum_{k=1}^{n}\alpha \left(x_{k}\right)$. Under such constraints, $\alpha \left(x_{i}\right)$ can quickly reach zero when $\theta_{i}$ is small. However, the objective functions may not be optimal at this time. Therefore, we could adjust $\theta_{i}$ according to the divergence degree. If $\alpha \left(x_{i}\right)$ is very large, the optimizable region can be narrowed; namely, by reducing $\theta_{i}$. This can be expressed as follows:(10)$$\theta_{i}\left(t+1\right)=\left\{\begin{array}{c} max\left\{\theta_{i}\left(t\right)-\theta_{i,min},\theta_{i,min}\right\}\;\;if\;\alpha_{r}\left(x_{i}\right)>1\\ \theta_{i}\left(t\right)\;\;\;\;\;\;\;\;if\;\alpha_{r}\left(x_{i}\right)=1\\ min\left\{\theta_{i}\left(t\right)+\theta_{i,min},\theta_{max}\right\}\;\;if\;\alpha_{r}\left(x_{i}\right)<1 \end{array}\right.,$$
where $\theta_{i}\left(t\right)$ is the control parameter of the *t* generation in the *i*th objective. For convenience, the initial value of $\theta_{i}$ is set to $\theta a_{i},a_{j}{}_{i,min}$, where j is the adjacent problem of the *i*th objective. $\theta_{i,max}$ is the double maximum angle between $a_{i}$ and m coordinates.

## 4. Test Results

In order to test the performance of the new hybrid algorithm, two types of test function were selected to compare the performances of the original MOEA/D, MOEA/ACD, MOEA/ ACD-NS, EN-MOEA/D, and EN-MOEA/ACD-NS. One type of test is the MOP test function set in CEC 2009 [31]. The other type of test is the unconstrained function (UF) problem, whose Pareto set is easy to be obtained [46]. A set of unconstrained functions proposed by Van Veldhuizen in 1999 can comprehensively test a multi-objective optimization algorithm, and is in accordance with the design criteria of the MOP test function proposed by Whitley et al. [46,47]. The evolutionary algorithm based on MOEA/D involves three kinds of decomposition methods, namely: the weighted sum approach (WS), the weighted Tchebycheff approach (WT), and the penalty-based boundary intersection approach (PBI). In this paper, we selected the PBI approach and generated new individuals by differential evolution. The mutation operator is a polynomial variation, and the weight vector is generated by the old method.

### 4.1. Results and Discussion of the Classical Test Functions

In this paper, we used test sets from Reference [48,49], which are UF1-UF9 and MOP1-MOP7, respectively. The search space dimensions of MOP1-MOP6 and UF1-UF6 are 2, and the search space dimensions of MOP7 and UF7-UF9 are 3. For the purpose of fairness, the recombination operators and decomposition methods of all of the algorithms in the comparative experiment are the same. In the programming implementation, we used a differential evolution operator [50] and polynomial mutation operator [48] as the recombination operators; furthermore, the method of the Chebyshev decomposition was used to decompose the multi-objective function. Considering the size of the population and the parameter of the neighbors’ number in ENS-MOEAD, we set the parameters NS in ENS-MOEAD, MOEA/ACD-NS, and EN-MOEA/ACD-NS to $\left\{10,20,25\right\}$.Through the experiment of independent parameters, we found that the range of $\theta_{i}$ within Reference [2,10] is better for the average experiment of the test sets UF1-UF9 and MOP1-MOP7. In this experiment, we set the value of $\theta_{i}$ as $\left\{3,5,7\right\}$. The general parameter settings for all MOP problems used for the testing are as follows:

Population size: 200 for MOP1 – MOP5, 150 for UF1-UF6, and the others are 240.

The number of independent runs: all test samples run 30 times independently.

Maximum function evaluation times: 15,000 times for all of the test samples.

Table 1, Table 2 and Table 3 show the mean value and standard deviation of HV, IGD, and the average Hausdorff distance ($\Delta_{2}$) in different test problems, where the optimal value is shown in bold. It can be seen from Table 1, Table 2 and Table 3 that, using HV, IGD, and $\Delta_{2}$ as metrics, the improved algorithms of MOEA/ACD, MOEA/ACD-NS, EN-MOEA/D, and EN-MOEA/ACD-NS obtain better evaluations compared with the original MOEA/D. In particular, the MOEA/ACD-NS performs better in most problems than other improved algorithms and the MOEA/D performs worst. Although the improved algorithm EN-MOEA/ACD-NS algorithm is inferior to MOEA/ACD-NS, it is no worse than EN-MOEA/D and MOEA/ACD in some problems. As both MOEA/ACD-NS and EN-MOEA/ACD-NS involve the choice of NS, we can speculate that the reason why other improved algorithms failed to obtain better metrics is that these algorithms may fall into local optima.

Figure 1 and Figure 2 show box plots of the metrics. The box plots contain the following of five values: maximum, minimum, the first quartile, the third quartile, and median. They show the shape, center, and dispersion of the data distribution. The “whiskers” above and below the rectangle show the positions of the minima and maxima, respectively, and the “+” in the figures indicate the outliers. In the MOPs, with HV and IGD as the evaluation indicators, the outliers of EN-MOEA/ACD-NS are significantly less than the other algorithms, indicating that our improved algorithm is more robust. Observing the quartiles and medians of the box plots, it is easy to see that EN-MOEA/ACD-NS is generally consistent with MOEA/ACD-NS except for MOP2 and MOP3 with IGD as the metric and the IGD of MOEA/ACD-NS is less than other algorithms, indicating that the two algorithms perform similarly in the calculation process and have a better accuracy. In the UFs, although the quartile and median of ENS-MOEA/ACD-NS differ from MOEA/ACD-NS, there are relatively few outliers in the various metrics of the hybrid algorithm, which further confirms the robustness of our algorithm. If IGD is taken as the metric, the median of EN-MOEA/ACD-NS is relatively small compared with the other algorithms, except for the problem UF10. Similarly, with HV as an indicator, the median of EN-MOEA/ACD-NS is larger than the other algorithms except for UF10. This result indicates that the EN-MOEA/ACD-NS has a better convergence and distribution. So, we can conclude that the hybrid algorithm performs better, as mentioned in the analysis of Table 1, Table 2 and Table 3.

### 4.2. Results and Discussion of the Joint Inversion of Atmospheric Ducts Problem

Today, with the development of technology, we can obtain the meteorological data by GNSS. In this paper, we placed a receiver on the ground and adjusted its antenna to a proper angle in order to receive the signal from the GNSS, and then recorded the GNSS phase delay and propagation loss observation in turn, which formed an automatic receiving sensor [30]. Considering the complexity of the atmospheric boundary layer, increasing the number of GNSS antenna sensors can effectively increase the information capacity of the atmospheric ducts, and then enhance the constraints of the retrieval of the atmospheric ducts’ parameters. Theoretically, it can improve the retrieval accuracy of the atmospheric ducts; therefore, we proposed the idea of obtaining GNSS signals from multiple sensors on the ground in order to jointly retrieve the atmospheric ducts. After we collected the data, we used the new algorithm to solve the practical problem of jointly retrieving atmospheric ducts. The specific framework of the joint retrieval of atmospheric ducts has been elaborated in our previous work [32,51]. This paper takes the problem as a series of practical test problems to verify whether the improved algorithm can be applied in practical conditions. According to the classification of this problem in Reference [32], fifteen types of problems in three situations were used. The maximum number of iterations was set to 50, and the population number was 300. The learning period (LP) was defined as 20. We ran the algorithm 20 times independently and analyzed the results statistically.

Table 4 shows the simulated values of the retrieved parameters, as well as the corresponding mean values and standard deviations. In order to show the results for diversity, we calculated the evaluation metric ∆_2_. Comparing the metric values, it is easy to find that the gap between the simulated and retrieved parameters gradually became larger when the number of retrieved parameters increased under the same known conditions. When the number of inversion parameters is the same, the deviation of individual parameters will increase when the known conditions increase. This is also shown by the metric ∆_2_. For example, when considering only one antenna to receive data, all ∆_2_ were less than 2, whereas, when three or four antennas were considered, the ∆_2_ in *D* = 5 and *D* = 6 were greater than 3. We can conclude that the convergence of the approximate solution set becomes weaker when the discretization degree of the decision space increases. Furthermore, the optimal solution obtained does not necessarily converge to the real value. In fact, similar conclusions can be reached from the results shown in Reference [51]. Therefore, we can conclude that increasing the known information may not effectively improve the retrieved effect in the actual operation process when the number of inversion parameters is certain.

Figure 3 shows the profiles retrieved and the corresponding absolute deviations versus heights in various problems. GPS1-5 represents the type of problem, and, in Table 4, *M* and *D* are the dimensions of the solution space and decision space, respectively. In GPS1, the retrieved profile obtained converges closely to the simulated profile, and the absolute deviation is less than 0.6N units. It could be claimed that the improved algorithm achieved better results. However, the absolute deviation is around 15N units or even 20N units in other cases, suggesting that the retrieved effect does not seem to be improved when the known information increases. After a careful comparison of the absolute deviation under the same *D* and different *M* conditions, we find that the retrieved parameters approximate to the real value. When the number of retrieved parameters *D* increased, the absolute deviation gradually increases between 100 and 400 m. The deviation between 400 and 600 m was slightly less than the deviation between 100 and 400 m in the same problem.

In order to better describe whether an increase in information content is effective for improving the retrieved results, we compared results for the retrieval of four parameters. As Figure 4 shows, the retrieved result is the best when the number of antennas is 1, and the worst when the number is 2. Moreover, the inversion effect does not seem to be improved as the number of antennas increases, which is consistent with the analysis of Figure 3.

## 5. Conclusions

To solve multi-objective optimization problems, we improved the original MOEA/D algorithm by dynamically adjusting the neighborhood sizes and improvement regions and proposed EN-MOEA/ACD-NS. The larger NS can enhance the global search ability. In order to properly adjust the scope of NS, we chose $p_{k,G}$ as the parameter to judge whether to change the NS. At the same time, the expansion of NS is bound to affect the convergence of the population. We can control parameter $\theta_{i}$ and dynamically adjust the convergence direction to improve the convergence accuracy and speed of our algorithm. To test its effectiveness and feasibility, the CEC 2009 standard test instances and the practical problem of jointly retrieving atmospheric ducts were used for numerical simulation experiments. The experimental results demonstrate that EN-MOEA/ACD-NS has a better computational performance, which is close to that of MOEA/ACD-NS, and it has a better convergence and distribution. When retrieving the refractivity parameters only, the improved algorithm can obtain a better retrieved value with the absolute deviation within 3N units. When the number of retrieved parameters increases, the value of ∆_2_ becomes larger and the diversity of the solution sets increases, which means that the difficulty of optimal solutions converging to the real values also increases. This new algorithm will provide a new method for the retrieval of atmospheric ducts, which is of great significance for the research of atmospheric ducts.

The performance of the proposed EN-MOEA/ACD-NS is better than other algorithms in the application of retrieving the atmospheric ducts. However, the NS and $\theta_{i}$ need to be adjusted by experiments according to practical problems. When the parameter evaluation for a real multi-objective problem is complicated, this algorithm will consume a lot of time. In the future, we will make a more detailed analysis on the parameter setting of EN-MOEA/ACD-NS and provide the reference parameter setting intervals under different decision space dimensions and search space dimensions to solve this problem.

## Figures and Tables

**Figure 1 sensors-20-02230-f001:**
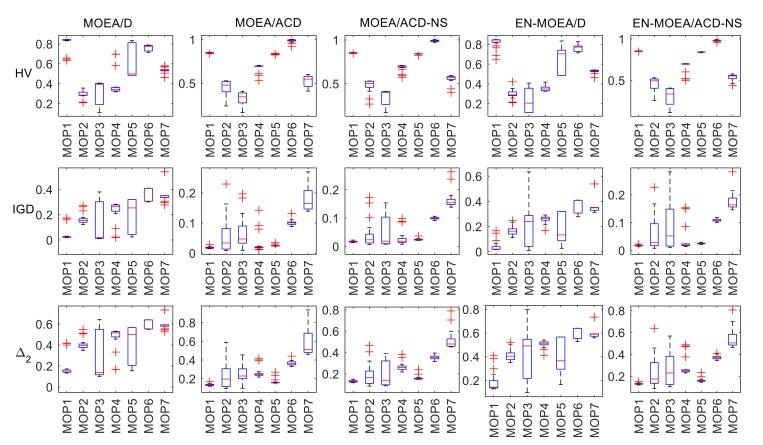
Box plots of the distribution of HV, IGD, and ∆_2_ values using five test algorithms in the MOPs.

**Figure 2 sensors-20-02230-f002:**
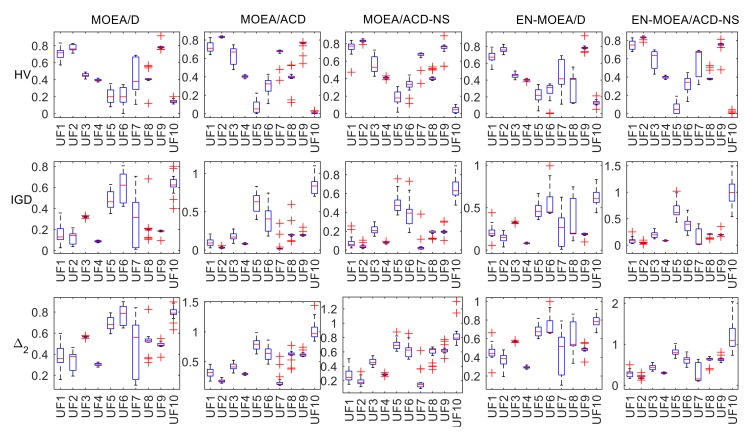
Box plots of the distribution of HV, IGD, and ∆_2_ values using five test algorithms in the UFs.

**Figure 3 sensors-20-02230-f003:**
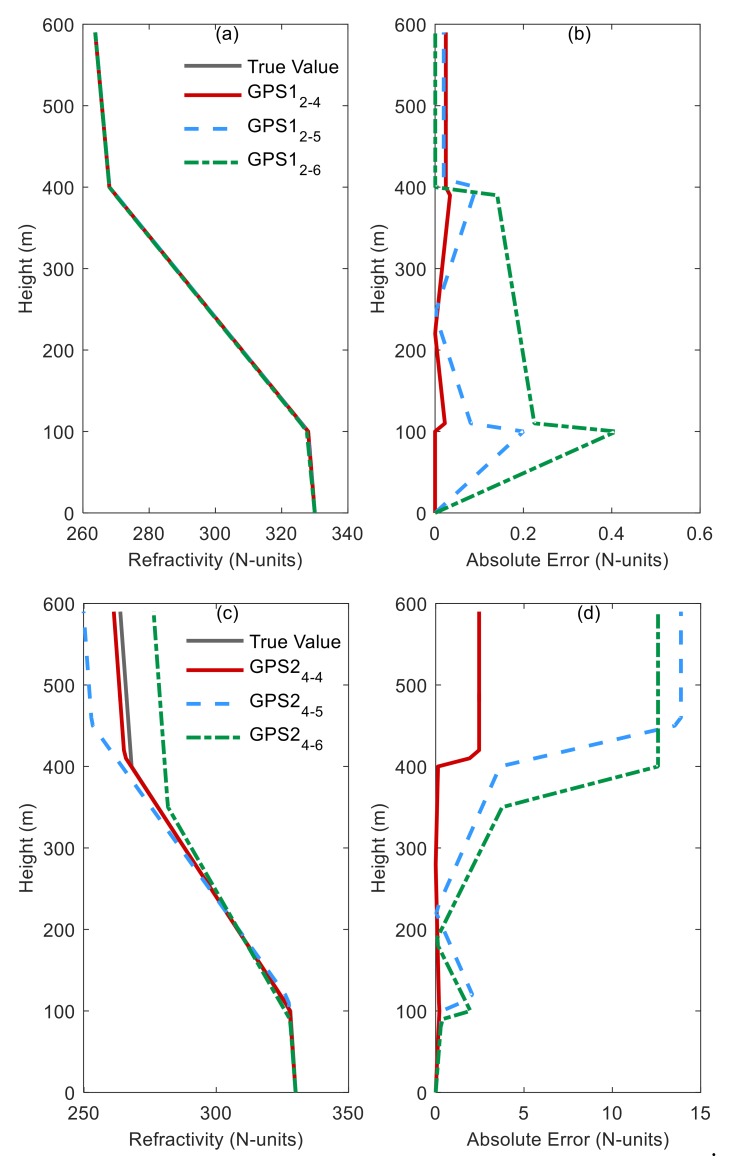
Comparison of simulated and retrieved profiles by EN-MOEA/ACD-NS in the problems for GPS1, GPS2, GPS3, GPS4, and GPS5, respectively. (**a**) is the simulated and retrieved profiles for GPS1; (**b**) is the absolute error of retrieved profiles for GPS1; (**c**) is the simulated and retrieved profiles for GPS2; (**d**) is the absolute error of retrieved profiles for GPS2; (**e**) is the simulated and retrieved profiles for GPS3 and GPS4; (**f**) is the absolute error of retrieved profiles for GPS3 and GPS4; (**g**) is the simulated and retrieved profiles for GPS5; (**h**) is the absolute error of retrieved profiles for GPS5.

**Figure 4 sensors-20-02230-f004:**
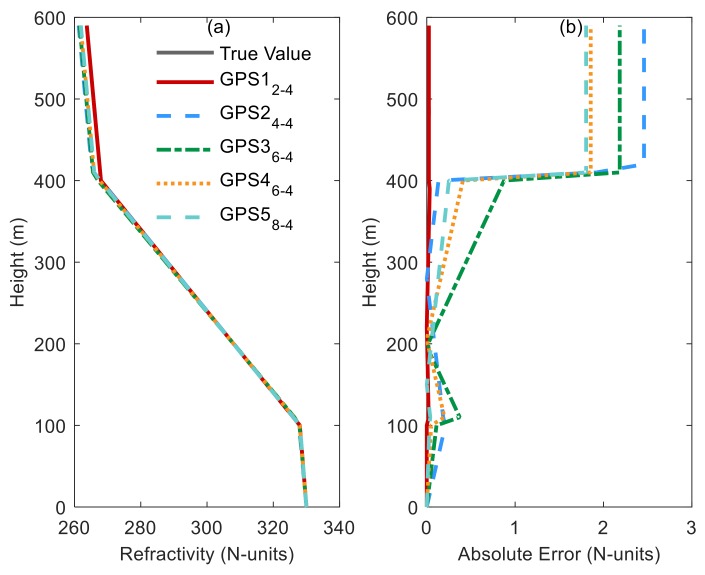
Comparison of different retrieved profiles by EN-MOEA/ACD-NS when considering the retrieval of four parameters. (**a**) is the simulated and retrieved profiles for different GPS problem; (**b**) is the absolute error of retrieved profiles for different GPS problem.

**Table 1 sensors-20-02230-t001:** Statistical results of hypervolume (HV) metric values obtained by decomposition-based multi-objective evolutionary algorithm (MOEA/D), MOEA/ACD, MOEA/ACD-NS, EN-MOEA/D, and EN-MOEA/ACD-NS (mean and SD). UF = unconstrained function; MOP = multi-objective optimization. The bold number is the optimal value of each set of functions.

Functions	Mean (SD)				
	MOEA/D	MOEA/ACD	MOEA/ACD-NS	EN-MOEA/D	EN-MOEA/ACD-NS
**UF1**	0.6999(0.0729)	0.7239(0.0562)	**0.7551(0.0790)**	0.6706(0.0629)	0.7509(0.0507)
**UF2**	0.7730(0.0386)	**0.8347(0.0077)**	0.8318(0.0156)	0.7629(0.0369)	0.8308(0.0144)
**UF3**	0.4534(0.0229)	**0.6260(0.0919)**	0.5630(0.0948)	0.4514(0.0267)	0.5933(0.0949)
**UF4**	0.3946(0.0133)	0.4033(0.0110)	**0.4090(0.0106)**	0.4036(0.0105)	0.3984(0.0124)
**UF5**	0.2068(0.0850)	0.0717(0.0694)	0.1816(0.0810)	**0.2173(0.0882)**	0.0589(0.0599)
**UF6**	0.1966(0.1090)	0.2960(0.0963)	0.3237(0.0744)	0.2681(0.1043)	**0.3242(0.0786)**
**UF7**	0.4473(0.2176)	**0.6540(0.0822)**	0.6502(0.0837)	0.4712(0.1772)	0.5806(0.1508)
**UF8**	0.4129(0.0876)	0.3838(0.0944)	**0.4177(0.0508)**	0.3154(0.1579)	0.3977(0.0468)
**UF9**	0.7864(0.0322)	0.7476(0.0577)	0.7443(0.0768)	**0.7871(0.0376)**	0.7434(0.0647)
**UF10**	**0.1440(0.0211)**	0.0079(0.0098)	0.0438(0.0332)	0.1298(0.0309)	0.0042(0.0111)
**MOP1**	0.8127(0.0722)	0.8509(0.0043)	**0.8524(0.0033)**	0.8173(0.0567)	0.8522(0.0024)
**MOP2**	0.2888(0.0417)	0.4545(0.0777)	0.4769(0.0713)	0.2971(0.0513)	**0.4560(0.0826)**
**MOP3**	0.3121(0.1088)	0.3261(0.0775)	**0.3440(0.0824)**	0.2381(0.1153)	0.3054(0.1105)
**MOP4**	0.3865(0.1200)	0.6792(0.0455)	**0.6808(0.0414)**	0.3520(0.0238)	0.6656(0.0697)
**MOP5**	0.6326(0.1657)	0.8391(0.0046)	**0.8404(0.0052)**	0.6293(0.1383)	0.8397(0.0033)
**MOP6**	0.7570(0.0295)	0.9858(0.0188)	**0.9900(0.0093)**	0.7612(0.0326)	0.9787(0.0092)
**MOP7**	0.5296(0.0330)	0.5294(0.0617)	**0.5536(0.0497)**	0.5258(0.0230)	0.5373(0.0376)

**Table 2 sensors-20-02230-t002:** Statistical results of inverted generational distance (IGD) metric values obtained by MOEA/D, MOEA/ACD, MOEA/ACD-NS, EN-MOEA/D, and EN-MOEA/ACD-NS (mean and SD). The bold number is the optimal value of each set of functions.

Functions	Mean (SD)				
	MOEA/D	MOEA/ACD	MOEA/ACD-NS	EN-MOEA/D	EN-MOEA/ACD-NS
**UF1**	0.1593(0.0835)	0.0983(0.0522)	**0.0801(0.0613)**	0.2173(0.0819)	0.0856(0.0549)
**UF2**	0.1224(0.0614)	**0.0317(0.0090)**	0.0376(0.0217)	0.1416(0.0572)	0.0346(0.0169)
**UF3**	0.3200(0.0063)	**0.1752(0.0502)**	0.2141(0.0441)	0.3235(0.0062)	0.1963(0.0514)
**UF4**	0.0864(0.0081)	0.0812(0.0072)	**0.0778(0.0069)**	0.0804(0.0068)	0.0842(0.0078)
**UF5**	0.4804(0.0864)	0.6216(0.1270)	0.4874(0.0968)	**0.4762(0.1002)**	0.6545(0.1489)
**UF6**	0.6001(0.1374)	0.4078(0.1773)	**0.3846(0.1399)**	0.5395(0.1644)	0.3950(0.1558)
**UF7**	0.2778(0.2576)	**0.0422(0.0841)**	0.0424(0.0831)	0.2368(0.1981)	0.1213(0.1612)
**UF8**	0.2094(0.1154)	0.2194(0.1021)	**0.1833(0.0323)**	0.3608(0.2347)	0.1959(0.0309)
**UF9**	**0.1814(0.0211)**	0.2002(0.0264)	0.1992(0.0411)	0.1845(0.0223)	0.2019(0.0365)
**UF10**	**0.6282(0.0903)**	0.8355(0.1020)	0.6488(0.1032)	0.6323(0.0972)	0.9867(0.2256)
**MOP1**	0.0436(0.0541)	0.0182(0.0032)	**0.0172(0.0024)**	0.0415(0.0436)	0.0173(0.0019)
**MOP2**	0.1660(0.0419)	0.0593(0.0613)	**0.0417(0.0474)**	0.1667(0.0381)	0.0600(0.0665)
**MOP3**	0.1395(0.1585)	0.0652(0.0543)	**0.0506(0.0515)**	0.2142(0.1790)	0.0850(0.0864)
**MOP4**	0.2288(0.0832)	0.0311(0.0335)	**0.0296(0.0275)**	0.2580(0.0286)	0.0419(0.0499)
**MOP5**	0.1904(0.1363)	0.0251(0.0028)	**0.0244(0.0034)**	0.1934(0.1200)	0.0250(0.0021)
**MOP6**	0.3521(0.0526)	0.1015(0.0089)	**0.0990(0.0038)**	0.3460(0.0530)	0.1087(0.0052)
**MOP7**	0.3653(0.0780)	0.1819(0.0424)	**0.1652(0.0313)**	0.3580(0.0636)	0.1773(0.0327)

**Table 3 sensors-20-02230-t003:** Statistical results of ∆_2_ metric values obtained by MOEA/D, MOEA/ACD, MOEA/ACD-NS, EN-MOEA/D, and EN-MOEA/ACD-NS (mean and SD). The bold number is the optimal value of each set of functions.

Functions	Mean (SD)				
	MOEA/D	MOEA/ACD	MOEA/ACD-NS	EN-MOEA/D	EN-MOEA/ACD-NS
**UF1**	0.3849(0.1081)	0.3128(0.0836)	**0.2723(0.0946)**	0.4583(0.0877)	0.2850(0.0861)
**UF2**	0.3373(0.0955)	**0.1766(0.0242)**	0.1877(0.0496)	0.3670(0.0857)	0.1822(0.0386)
**UF3**	0.5656(0.0056)	**0.4158(0.0584)**	0.4604(0.0473)	0.5688(0.0054)	0.4394(0.0584)
**UF4**	0.3032(0.0141)	0.2935(0.0120)	**0.2868(0.0121)**	0.2924(0.0115)	0.2990(0.0135)
**UF5**	0.6905(0.0619)	0.7902(0.0913)	0.7020(0.0730)	**0.6867(0.0702)**	0.8131(0.0975)
**UF6**	0.7698(0.0894)	0.6405(0.1223)	**0.6183(0.0997)**	0.7277(0.1025)	0.6213(0.1203)
**UF7**	0.4494(0.2826)	**0.1739(0.1259)**	0.1818(0.1270)	0.4332(0.2276)	0.2764(0.2182)
**UF8**	**0.5201(0.0999)**	0.6114(0.0888)	0.5866(0.0998)	0.6153(0.1617)	0.6106(0.0988)
**UF9**	0.4886(0.0335)	0.6215(0.0399)	0.6274(0.0580)	**0.4863(0.0395)**	0.6353(0.0550)
**UF10**	**0.7905(0.0586)**	1.0085(0.1498)	0.8356(0.1440)	0.7929(0.0609)	1.1933(0.3305)
**MOP1**	0.1859(0.0974)	0.1345(0.0114)	**0.1308(0.0089)**	0.1863(0.0845)	0.1314(0.0071)
**MOP2**	0.4073(0.0537)	0.2351(0.1347)	**0.1907(0.1066)**	0.4107(0.0512)	0.2344(0.1516)
**MOP3**	0.3052(0.2278)	0.2451(0.1033)	**0.1966(0.1139)**	0.4235(0.2232)	0.2670(0.1464)
**MOP4**	0.4674(0.1119)	**0.2680(0.0605)**	0.2722(0.0470)	0.5071(0.0298)	0.2870(0.0882)
**MOP5**	0.4005(0.1777)	0.1683(0.0305)	**0.1619(0.0210)**	0.4156(0.1476)	0.1630(0.0198)
**MOP6**	0.5919(0.0441)	0.3658(0.0255)	**0.3546(0.0204)**	0.5866(0.0446)	0.3720(0.0166)
**MOP7**	0.6015(0.0605)	0.5935(0.1536)	**0.5146(0.0892)**	0.5965(0.0488)	0.5452(0.0885)

**Table 4 sensors-20-02230-t004:** Statistical results of the retrieved parameter values and ∆_2_ metric in different problems (mean and SD).

Problems	*M*	*D*	Inversion Slope *c*_1_	Height *h*_1_	Inversion Slope *c*_2_	Height *h*_2_	Δ_2_
**Simulated** **Results**	-	-	100	300	−0.02	−0.2	-
**GPS1**	2	4	100.1342 (0.0065)	299.8013 (0.0060)	0.02 (0.0000)	0.2002 (0.000)	1.666 (0.1924)
**GPS1**	2	5	100.6387 (0.0091)	299.7782 (0.0057)	0.0220 (0.0000)	0.1994 (0.0059)	1.6453 (0.0965)
**GPS1**	2	6	101.0383 (0.0802)	298.1824 (0.1055)	0.0241 (0.0002)	0.1997 (0.0002)	1.7403 (0.1496)
**GPS2**	4	4	102.2753 (3.2614)	310.7474 (20.0969)	0.0221 (0.0018)	0.2011 (0.0028)	1.9413 (0.0727)
**GPS2**	4	5	114.3822 (12.0344)	337.3833 (14.1259)	0.0235 (0.0033)	0.2204 (0.0163)	2.8771 (0.0826)
**GPS2**	4	6	89.4901 (10.8181)	260.8328 (12.1285)	0.0235 (0.0047)	0.1771 (0.0209)	2.8869 (0.0657)
**GPS3**	6	4	102.8699 (10.8528)	304.3318 (13.2847)	0.0211 (0.0036)	0.2043 (0.0165)	2.2986 (0.0693)
**GPS3**	6	5	99.0262 (17.7310)	257.8824 (8.6933)	0.0354 (0.0190)	0.1482 (0.0649)	3.3573 (0.0729)
**GPS3**	6	6	112.3220 (16.0652)	348.4416 (4.0524)	0.0329 (0.0070)	0.2170 (0.0289)	3.7982 (0.1053)
**GPS4**	6	4	101.4928 (5.7184)	306.5612 (15.5952)	0.0205 (0.0033)	0.2021 (0.0070)	2.3387 (0.0482)
**GPS4**	6	5	124.9876 (14.0261)	347.9198 (2.9298)	0.0261 (0.0048)	0.2334 (0.0215)	3.3931 (0.0573)
**GPS4**	6	6	89.6128 (13.9242)	256.1806 (7.1527)	0.0265 (0.0062)	0.1661 (0.0440)	3.3632 (0.0622)
**GPS5**	8	4	100.1070 (0.1295)	308.5910 (17.9959)	0.0197 (0.0011)	0.2010 (0.0017)	2.5372 (0.0503)
**GPS5**	8	5	123.2109 (18.3929)	335.7955 (28.5190)	0.0253 (0.0038)	0.2321 (0.0409)	3.8360 (0.0680)
**GPS5**	8	6	122.6695 (16.0692)	344.6790 (5.0537)	0.0292 (0.0074)	0.2304 (0.0218)	3.7798 (0.0850)

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
