# Peer review of "Using the Decomposition-Based Multi-Objective Evolutionary Algorithm with Adaptive Neighborhood Sizes and Dynamic Constraint Strategies to Retrieve Atmospheric Ducts"

_sensors, 2020, doi:10.3390/s20082230_

Round 1

Reviewer 1 Report

"The manuscript entitled "Decomposition-Based Multiobjective Evolutionary Algorithm with Adaptive Neighborhood Sizes and Dynamic Constraint Strategies ", by Yanbo Mai, Hanqing Shi, Qixiang Liao, Zheng Sheng, Shuai Zhao, Qingjian Ni, and Wei Zhang presents an interesting work.

In general, the manuscript should be acceptable for publication but some minor problems should be repaired prior to publication. Some suggestions are as follows:

(1) English language, grammar, and technical errors. There are several English language and technical errors in the paper. Also, there are some incorrect sentences.

(2) In the conclusion section, the authors should provide more information about their future work. For example, what is the potential of the proposed method?

(3) the introduction should be extended to include the research question, objective(s) and goal(s) of the research and research methodology, possibly as a separate subsection within the Introduction.

(4) The full form of all of the abbreviations must be added to the manuscript.

(5) The reference list should be expanded. The references should be updated with newer once from the year 2019 as well as the year 2020.

Good Luck!

Reviewer 2 Report

In this paper, the authors propose a new variant of a decomposition based MOEA for the treatment of multi-objective optimization algorithms. The method is tested on some benchmark problems and on some problems related
to atmospheric ducts.

Overall I think the paper can be published after some modifications in Sensors. My main concern is the choice of this journal. The main contribution of this paper is the development of a novel MOEA, and the application to GNS problems is only treated marginally. This, however, is to the editors to decide.

In the following I list my comments on specific issues I have found when reading the manuscript, and which may be addressed by the authors for their revision, in case the paper should be published in Sensors.

- the English is ok and I could read the manuscript, but for sake of a better readability the English should undergo a revision. Also, some inconsistencies should be removed. Not too important, but the paper has some abuse of notations. For instance, m is used for the number of objectives, later for the population size and for the size of the PF discretization.

- intro: the discussion on the mathematical programming techniques is way too short and unfair as there is up to date much more than scalarization methods (which, however, still represent a huge class of MP methods for the treatment of MOPs). There are, for instance, cell to cell mapping methods, subdivision techniques, or continuation-like
strategies that all have their advantages (and, of course, disadvantages).

- MOO: the description of multi-objective optimization could be improved. For instance, dominance is slightly wrong, as in addition it is required that there exists an index l such that f_l(x_a) is strictly less "<" than f_l(x_b) (else, you run into troubles for points x_a and x_b that have the same objective values, F(x_a) = F(x_b)
Further, I do not like the notation "sub-problem" for that is commonly termed  "objective". This implies that there are "only" m problems to be solved separately, while the PS typically forms an objective of dimension m-1.
In page 3, on top you write that "sub-problems cannot be optimized synchronously" Well, THAT is the idea of MOO.

- Sec. 4.2: I personally would like to see a better description and more in-depth discussion on the GPS problems, as they are the reason why you place the article in Sensors.  Else, I suggest to chose another journal.

- the first sentence of the abstract does not seem to be complete

Pareto Tracer: a predictor–corrector method for multi-objective optimization problems Adanay Martín and Oliver Schtze. Engineering Optimization, 50(3): 516-536, 2018

Reviewer 3 Report

This paper uses multi-objective optimization to solve the problem of retrieving atmospheric ducts with GNSS signals, ..
This paper proposes a hybrid decomposition-based multi-objective evolutionary algorithm with adaptive neighborhood sises which imposes dynamically some constraints on the sub-problems.

The paper is well written, it is easy to follow, the Tables and Figures are acceptable and the results sound achievable. However, I consider that the following points would help improve the paper:
1. It would be important for the authors to highlight the problem they are trying to solve. Describing the importance of using multi-objective optimization to solve the problem.
2. It is necessary that the authors highlight the results obtained and better explain the Tables and Figures 1 and 2 of the results obtained.
3. The authors should deepen the analysis in the selection of their parameters
4. The authors should highlight and deepen their final conclusions
.

Reviewer 4 Report

The authors study a hybrid decomposition-based multi-objective evolutionary algorithm with adaptive neighborhood sizes, which imposes dynamically some constraints on the sub-problems.
The examined topic is of great interest and applicability. The provided analysis in is concrete and correct.
The paper is overall well written and easy to follow. The authors provide extensive numerical and experimental results, thus studying in detail the proposed problem.
The manuscript needs a minor revision before acceptance.

Round 2

Reviewer 2 Report

The authors have addressed all my comments, I am happy with the paper.

Author Response

Thank you for your comments, and wish you have a good life.